# Investigation of the incidence trend of follicular lymphoma from 2008 to 2017 in Taiwan and the United States using population-based data

Yu-Chieh Su[1,2], Brian Chih-Hung Chiu[3], Hung-Ju Li[1], Wen-Chi Yang[1,2], Tsai-Yun Chen[4], Su-Peng Yeh[5,6], Ming-Chung Wang[7], Wen-Tsung Huang[8], Ming-Yang Lee[9], Sheng-Fung Lin[1] *

1 Division of Hematology-Oncology, Department of Internal Medicine, E-Da Hospital, Kaohsiung, Taiwan, 2 School of Medicine, College of Medicine, I-Shou University, Kaohsiung, Taiwan, 3 Department of Public Health Sciences, The University of Chicago, Chicago, Illinois, United States of America, 4 Division of Hematology/Oncology, Department of Internal Medicine, National Cheng Kung University Hospital, Tainan, Taiwan, 5 Division of Hematology and Oncology, Department of Internal Medicine, China Medical University Hospital, Taichung, Taiwan, 6 College of Medicine, China Medical University, Taichung, Taiwan, 7 Division of Hema-Oncology, Department of Internal Medicine, Kaohsiung Chang Gung Memorial Hospital, Kaohsiung, Taiwan, 8 Division of Hematology and Oncology, Department of Internal Medicine, Chi Mei Medical Center, Liouying, Tainan, Taiwan, 9 Division of Hematology and Oncology, Department of Medicine, Chia-Yi Christian Hospital, Chia-Yi, Taiwan

* shlintw@yahoo.com.tw

## Abstract

### Background

The incidence of follicular lymphoma (FL) in Taiwan has not been well investigated since its inclusion as a histological subtype in the Taiwan Cancer Registry in 2008. The purpose of this study was to describe the incidence patterns of FL in Taiwan and compare the trends with those in other racial groups in the United States.

### Materials and methods

We conducted an epidemiological study using population-based data from the Taiwan Cancer Registry, Ministry of Health and Welfare, and the 18 Surveillance, Epidemiology, and End Results (SEER) registries to evaluate the FL incidence from 2008 to 2017. We calculated the annual percent change (APC) to describe the trends in the incidence of FL in subpopulations defined by race and sex over time.

### Results

The annual age-adjusted incidence rate of FL in Taiwan increased significantly from 0.59 per 100,000 persons in 2008 to 0.82 per 100,000 persons in 2017, with an APC of 3.2. By contrast, the incidence rate in whites in the United States during the same period decreased from 3.42 to 2.74 per 100,000 persons, with an APC of −2.1. We found no significant change for the blacks (APC, −1.5%), Hispanics (APC, −0.7%), and Asians or Pacific Islanders (APC, +0.7%). The temporal trend was similar between the males and females. The relative

**Funding:** This work was supported by grant EDAHP110021 from E-Da Hospital, Taiwan. The funder had no role in study design, data collection and analysis, decision to publish, or preparation of the manuscript.

**Competing interests:** The authors have declared that no competing interests exist.

frequency of FL among the incident non-Hodgkin lymphoma (NHL) cases also increased significantly in Taiwan from 7.64% in 2008 to 11.11% in 2017 (APC = 3.8). The relative frequency of FL among the incident NHL cases in the whites decreased from 2008 to 2012 (APC, −3.8%) and then stabilized after 2012 (APC, −0.2%). By contrast, little change in relative frequency of FL among the incident NHL cases was observed in the blacks, Hispanics, and APIs between 2008 and 2017.

## Conclusion

We found increases in the incidence of FL and the relative frequency of FL among the incident NHL cases in both males and females in Taiwan from 2008 to 2017. The FL incidence rates were unchanged for all races and sex groups in the United States, except for the decreases in the whites.

## Introduction

Follicular lymphoma (FL) is one of the most common subtypes of non-Hodgkin lymphoma (NHL) in the United States (US), with an incidence rate of 3·5 per 100,000 persons per year in 2008–2017 and representing 12·4% of all mature NHLs [1]. FL accounted for a higher percentage of NHLs in the United States and Western countries than in Asia [2–5]. However, evidence is accumulating that the FL incidence rate has been increasing in certain Asian countries. The proportion of FLs in Japan increased from 18·3% in 2000–2006 to 22·4% in 2007–2014 [6]. Chihara et al. showed that the world age-standardized incidence rates of FL in Japan increased from 0·1 to 1·1 per 100,000 persons between 1993 and 2008 [7]. A similar upward trend was also observed in South Korea during the 1999–2015 period [8, 9] (from 0·2 to 0·6 per 100,000 persons) and in Taiwan from 1990 to 2012 (from 0·3 to 0·9 per 100,000 persons) [10, 11].

Intensive epidemiological research, including those from the International Lymphoma Epidemiology Consortium (InterLymph), has provided intriguing new insights into the possible risk factors of FL, such as anthropometrics, family history of NHL, genetic susceptibility, pesticide use, and alcohol use [4, 12, 13]. However, the geographic heterogeneity of the incidence of FL and the upward trend in some Asian countries are widely recognized but remain largely unexplained. In addition, the increasing proportion of FLs in NHLs, approaching to that in Western countries in some regions [7, 14] but not in other regions [15, 16] in Asia remains incompletely understood. Additional evaluation of the incidence patterns in Asia using population-based registries is needed to provide leads and information for formulating etiological hypotheses.

We therefore analyzed Taiwanese and US population-based data on FL from the Taiwan Cancer Registry (TCR) and the US National Cancer Institute's Surveillance, Epidemiology, and End Results (SEER) Program, respectively. As FL was not reported as a subtype in the TCR until 2008 [17], we limited our analysis to the period from 2008 to 2017, the most recent period with available complete data.

## Materials and methods

### Study purpose

The objective of this study was to describe the incidence patterns of FL in Taiwan and compare the trends with those in other racial groups in the US.

## Patient population

We obtained FL cases in Taiwan for the period 2008–2017 from the TCR annual report of the Health Promotion Administration, Ministry of Health and Welfare, Taiwan [17] (S1 Table in S1 File). The data of FL cases in US for the study period of 2008–2017 was using the US National Cancer Institute's SEER 18 program for the period 2008–2017 [18] (S1 Table in S1 File).

## Data quality and completeness

The annual population was obtained from the Department of Household Registration, Ministry of the Interior, Taiwan [19] (S1 Table in S1 File). The cancer registry annual report based on the databased established by the TCR which case completeness rates increased from 97·6% in 2008 to 98·3% in 2017 [17, 20]. The TCR implemented rigorous quality control protocols to validate and verify cancer diagnoses, including submission of tissue pathology reports and random sampling of claims and diagnosis reports. It is considered a complete and accurate registry [21]. All information on the primary cancer site and histology was coded in accordance with the third edition of the International Classification of Diseases for Oncology (ICD-O-3). FL was added as a distinct lymphoma subtype to the TCR in 2008, in accordance with the 2008 revision of the World Health Organization classification [17]. Other routinely reported common subtype categories include diffuse large B-cell lymphoma, marginal zone lymphoma, peripheral T-cell lymphoma, and extranodal NK/T-cell lymphoma [17].

The SEER registries covered approximately 35% of the US population. For each newly identified case, SEER registries report patient demographic data, including age, race, ethnicity, and sex, and information on the tumor histological type, primary site, and stage at diagnosis. The SEER program records race as assigned by the North American Association of Central Cancer Registries, which is only comprehensive source of population-based in the US and have a case completeness rate greater than 98% [22].

## Data analysis

The SEER racial groups included non-Hispanic whites (hereafter referred to as whites), Hispanic whites (hereafter referred to as Hispanics), non-Hispanic blacks (hereafter referred to as blacks), and Asians/Pacific Islanders (APIs). Owing to the low numbers of cases, we did not include American Indians/Alaskan Natives in the present analysis.

Patients with FLs were identified using the *International Classification of Disease for Oncology, Third Edition* (*ICD-O-3*; Histology codes: 9690, 9691, 9695, and 9698) [23]. The incidence rates and relative frequencies of FL were calculated for FL overall and according to sex and racial groups as follows: Taiwanese (TWs), whites, Hispanics, blacks, and APIs. The age-adjusted incidence rates were standardized to the 2000 world standard population [24].

The incidence rates and relative frequency of FL among the incident NHL cases were calculated using SPSS version 24.0 (released 2016, IBM SPSS Statistics for Windows, Version 24.0, IBM Corp., Armonk, NY) and SEER*Stat software, Version 8.3.8 (Surveillance Research Program, National Cancer Institute) [25]. Temporal trends for the age-adjusted incidence rates and relative frequency of FL among the incident NHL cases from 2008 to 2017 were characterized by the annual percent change (APC) and 95% confidence intervals by using the weighted least squares method in the Joinpoint Regression Program, Version 4.8.0.1 (NCI Statistical Methodology and Applications Branch, Bethesda, MD) [26, 27]. APC is often used to characterize trends in disease rates, which assume the disease rates change at a constant percentage of the rate of the previous year by using this approach. The Joinpoint Regression Program described the change of data trend by several connected line segments on a logarithmic scale

with turning points or "joinpoints." The program used the permutation test to find the number of significant joinpoints. If no joinpoint was detected, it presented the data trend as a straight line with a single APC. When two or more joinpoints were detected, the program showed the corresponding APC for each line segment (time period) and the year of the joinpoint. The trend between males and females was compared by using a test of parallelism. An independent two-sided $t$ test was used to determine if the APC was statistically significant from 0. The statistical significance of the differences was assessed at a $p$ value $< 0.05$.

## Results

From 2008 to 2017, 2,494 FL cases were reported to the TCR in Taiwan. Of the patients, 1,279 (51·3%) were males. During the same period, 27,611 patients with FL were reported to the 18 SEER registries, including 24,517 whites, 1,494 blacks, 3,956 Hispanics, and 1,600 APIs, of whom 12,486 (50·9%), 716 (47·9%), 1,848 (46·7%), and 826 (51·6%) were males, respectively.

 Figs 1 and 2 show the world age-adjusted incidence rates of FL and the relative frequency of FL among the incident NHL cases for FL overall and stratified by sex according to race. The APCs and 95% confidence intervals (CIs) are listed in Table 1. The incidence of FL increased significantly from 0·59 to 0·82 per 100,000 person-years between 2008 and 2017 in Taiwan, representing a 39% increase and an APC of 3·2% (95% CI: 1·6%–4·8%). The upward trend in FL incidence occurred in both the males and females. The incidence rate increased by 2·6% (95% CI: 0·3%–5·0%) per year from 0·64 to 0·81 cases per 100,000 person-years in the males and by 3·8% (95% CI: 1·4%–6·3%) per year from 0·54 to 0·85 cases per 100,000 person-years in the females. By contrast, the incidence rates declined by −2·1% (95% CI: −2·9% to −1·2%) per year in the whites, from 3·42 to 2·74 per 100,000 person-years between 2008 and 2017, representing a 20% decrease. For the white males and females, the incidence rates decreased significantly by −1·8% (95% CI: −2·9% to −0·7%) and −2·4% (95% CI: −3·1% to −1·7%) per year from 3·75 to 3·00 per 100,000 person-years and from 3·11 to 2·50 per 100,000 person-years, respectively. The tests of parallelism for the male and female trend lines in the Taiwanese population and US white population both were not significant ($p$ = 0.456 and 0.133). During the same period, no significant changes in the age-adjusted incidence rates were found in the blacks, Hispanics, and APIs, regardless of sex, as shown in Table 1.

 Fig 2 and Table 1 show the trends in the relative frequency of FL among the incident NHL cases. In Taiwan, the relative frequency of FL among the incident NHL cases increased from 7·6% to 11·1% from 2008 to 2017, with an APC of 3·8% (95% CI: 0·9%–6·8%). The relative frequency of FL among the incident NHL cases increased in both the males and females, from 7·1% to 9·3% between 2008 and 2017 (APC, +3·4%, 95% CI: 0·1%–6·8%) in the males and from 8·5% to 13·4% (APC, +4·2%, 95% CI: 0·6%–7·9%) in the females. In the whites, the relative frequency of FL among the incident NHL cases declined from 12.9% to 10.7%, the average APC was −1.8 (95% CI: −3.1% to −0.5%, $p$ = 0.009). In the white males and females, the relative frequency of FL among the incident NHL cases declined from 11·6% to 9·6% (APC = −1·6, 95% CI: −2·6% to −0·6%) and from 14·6% to 12·2% (APC = −1·9, 95% CI: −2·8% to −0·9%), respectively. By contrast, the relative frequency of FL among the incident NHL cases remained stable during 2008–2017 among the blacks, Hispanics, and APIs, regardless of sex (Table 1).

## Discussion

To the best of our knowledge, this report presents the most recent temporal trend of the incidence of FL in Taiwan. We found that the incidence of FL and the relative frequency of FL among the incident NHL cases increased significantly between 2008 and 2017 in Taiwan, and the upward trend was observed in both males and females. By contrast, during the same

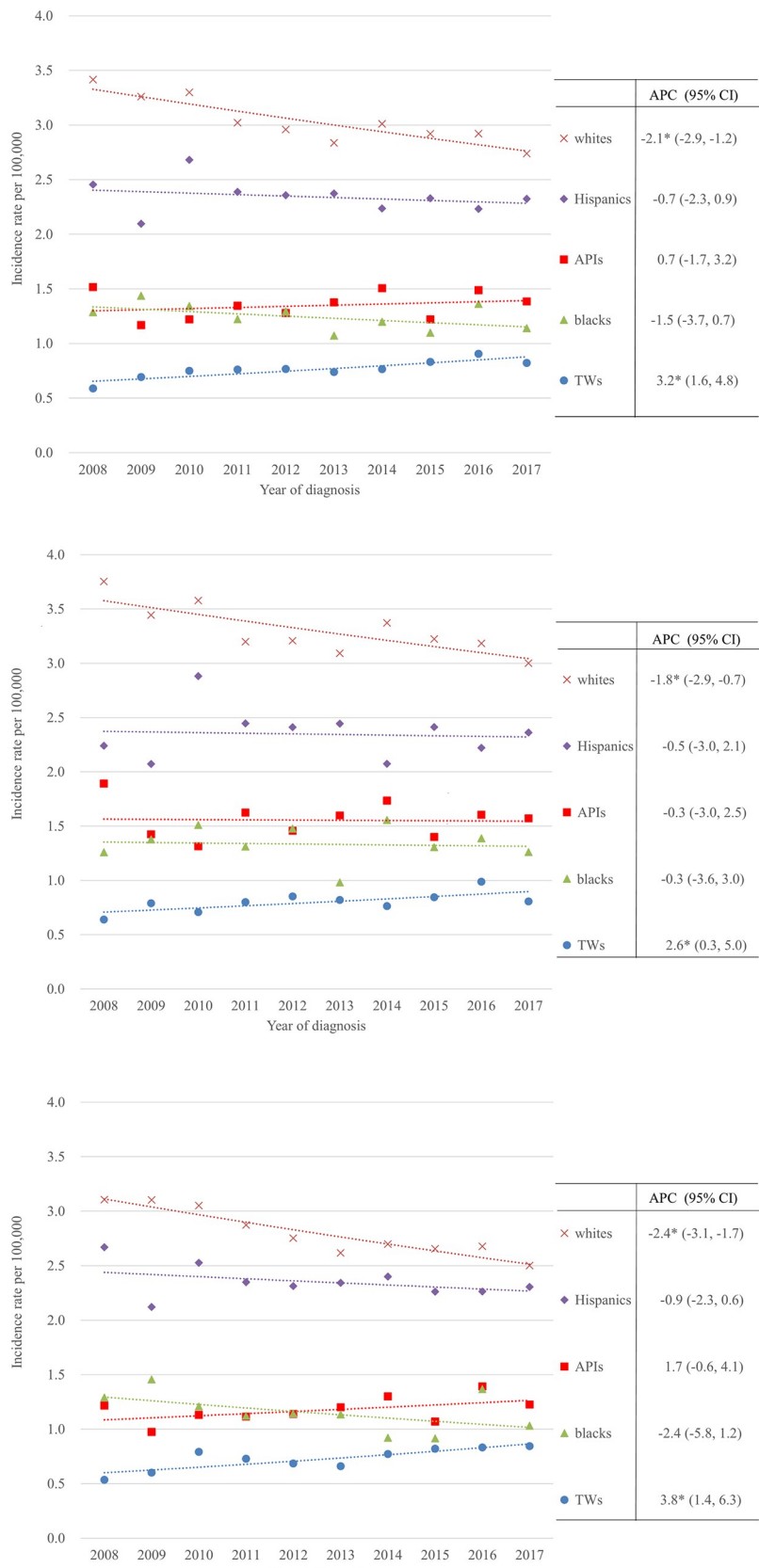

**Fig 1. Trends in the world age-adjusted incidence rate according to racial group for the period 2008–2017.** (A) Overall. (B) Males. (C) Females. *The APC is significantly different from 0 at the alpha level of 0.05.

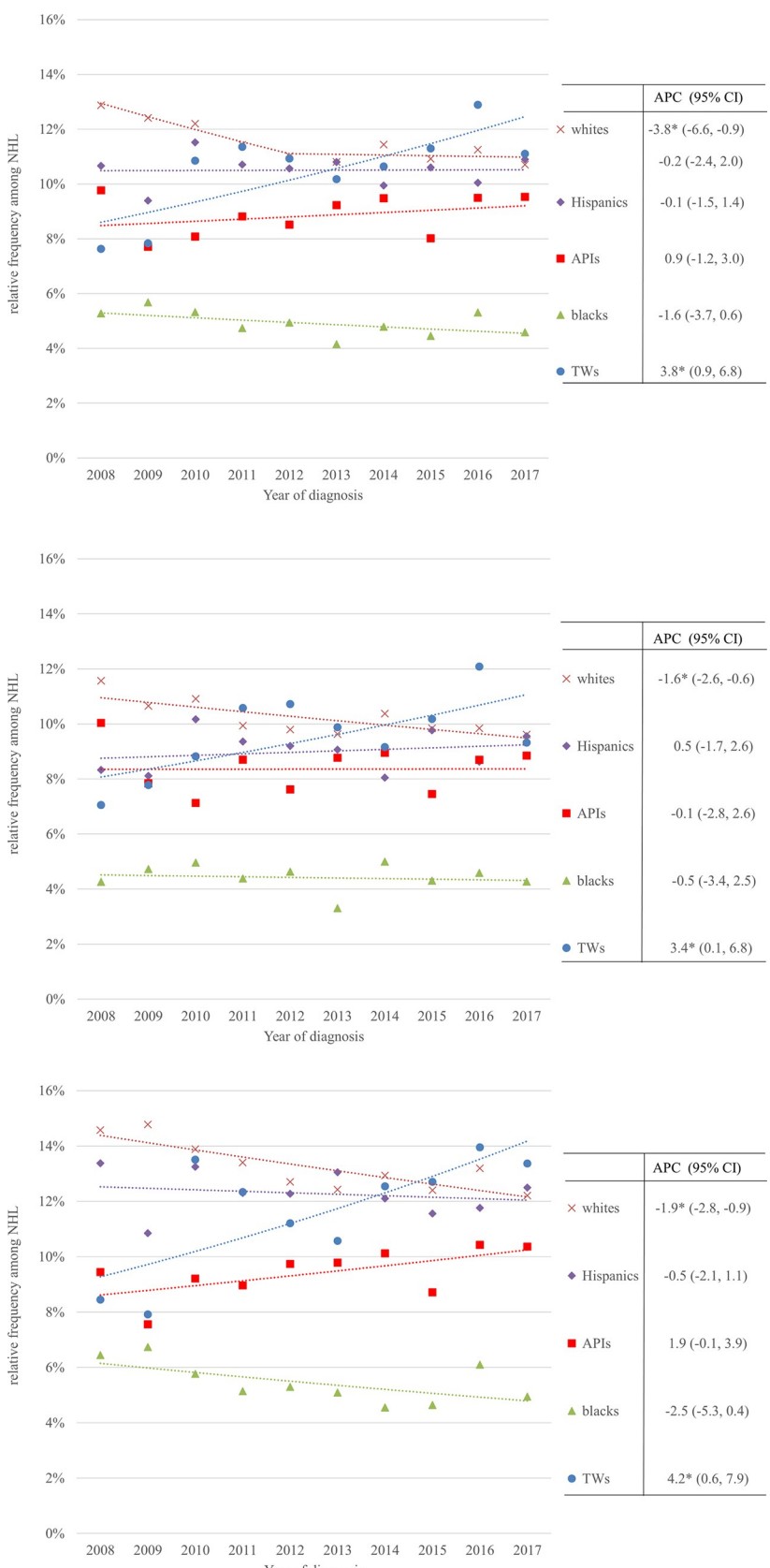

**Fig 2. Trends in the relative frequency of FL among the incident NHL cases according to racial groups for the period 2008–2017.** (A) Overall. (B) Males. (C) Females. *The APC is significantly different from 0 at the alpha level of 0.05.

period, the FL incidence rate decreased in the whites and appeared to remain stable in the blacks, Hispanics, and APIs. The reasons for these differences have not been well explained to date.

The leading causes of the increasing incidence of FL in Taiwan are largely unknown. Several risk factors of FL have been suggested, including sedentary lifestyle, obesity, diet, environmental exposures, and recreational sun exposure [4, 13, 28]. Part of the lower risk of FL in Asians may be genetics [29], and several genetic loci have been identified, particularly in the human leukocyte antigen (HLA) region [30]. Genetic variants, however, are unlikely to fully explain the diverse trend of FL in different racial groups. Our findings of a significant increase in the incidence of FL from 2008 to 2017 in Taiwan suggest the importance of modifiable lifestyle and environmental factors. For example, cigarette smoking has been associated with the risk of FL in two large InterLymph pooled analyses [13, 31]. The frequency of $t(14;18)$ translocation, which occurs in approximately 85% of FLs, is 3·6-fold higher in heavy smokers (>40 pack-years) than in nonsmokers and may explain the association between smoking and FL. However, a case-control study found no association between smoking and $t(14;18)$-positive NHL in men and women [32]. The dietary patterns in Taiwan are becoming increasingly westernized, featuring high meat and low vegetable consumptions. Meat intake and the components in meats (e.g., fat) have been associated with a risk of FL [33–38]. Dietary patterns high in meat and fat contents have been linked to excess FL risk in a case-control study [39], but not in two large prospective cohorts [34]. Obesity, a potential risk factor of FL [13], may also explain the upward trend of FL in Taiwan, as the prevalence of obesity has been increasing from 4·1% (1993–1996) to 6·2% (2005–2008), reaching 8·2% in 2013–2014 in Taiwan [40]. Recreational sunlight exposure was inversely associated with FL risk [13]. A recent study in Taiwan found that vitamin D deficiency in men and women was partly due to lack of sun exposure [41, 42]. Further studies are needed to evaluate the association between sun exposure and the development of FL in Taiwan. A combination of these modifiable risk factors may have contributed to the increasing FL incidence in Taiwan. Although the incidence of FL showed no significant changes in the APIs in the US, it was found to be increasing in other Asian countries such as Japan and South Korea and appeared to be increasing more in South Korea [8, 9] and Japan [7] than in Taiwan (S2 Table, S3 Table in S1 File and S1 Fig). The reasons for the increasing incidence of FL in these three Asian countries may be similar, but further research is needed.

For the incidence rates in Taiwan from 2002 to 2017, no joinpoint was detected, which suggests a steady increase in incidence over this period. The incidence rates in males and females in Taiwan continue to increase with APCs of 2·3 (95% CI: 1·3–3·4) and 4·1 (95% CI: 3·0–5·3), respectively (S3 Table in S1 File). The latest data indicated that the FL incidence rates in 2018 in Taiwan were 0·95 per 100,000 person-years for males and 1·00 per 100,000 person-years for females [17], which were within the 95% CI of APC calculated using the data from 2008 to 2017. These data suggest that the FL incidence in Taiwan may continue to increase in the near future.

The main limitation of this study is that SEER data does not cover whole US population, the incidence change in each racial population in US cannot be generalized. Another limitation of our study is that the FL data in Taiwan was retrieved from the TCR annual report, not Taiwan Cancer Registry Database. The number of FL cases by gender and age in the TCR

**Table 1. Number of follicular lymphoma cases and trends in age-adjusted rates and relative frequency of FL among the incident NHL cases according to sex for the period 2008–2017.**

| | | Total | 2008 | 2009 | 2010 | 2011 | Year 2012 | 2013 | 2014 | 2015 | 2016 | 2017 | APC (95% CI) | p |
|---|---|---|---|---|---|---|---|---|---|---|---|---|---|---|
| **TWs** | | | | | | | | | | | | | | |
| Total | Cases | 2494 | 169 | 205 | 226 | 238 | 249 | 239 | 263 | 290 | 319 | 296 | | |
| | IR | 0.77 | 0.59 | 0.69 | 0.75 | 0.76 | 0.77 | 0.74 | 0.77 | 0.83 | 0.91 | 0.82 | 3.2* (1.6, 4.8) | 0.002 |
| | RF (%) | 10.5 | 7.6 | 7.8 | 10.9 | 11.4 | 10.9 | 10.2 | 10.6 | 11.3 | 12.9 | 11.1 | 3.8* (0.9, 6.8) | 0.017 |
| Males | Cases | 1279 | 91 | 115 | 104 | 123 | 136 | 130 | 127 | 145 | 169 | 139 | | |
| | IR | 0.81 | 0.64 | 0.79 | 0.71 | 0.80 | 0.85 | 0.82 | 0.76 | 0.85 | 0.99 | 0.81 | 2.6* (0.3, 5.0) | 0.032 |
| | RF (%) | 9.5 | 7.1 | 7.8 | 8.8 | 10.6 | 10.7 | 9.9 | 9.2 | 10.2 | 12.1 | 9.3 | 3.4* (0.1, 6.8) | 0.047 |
| Females | Cases | 1215 | 78 | 90 | 122 | 115 | 113 | 109 | 136 | 145 | 150 | 157 | | |
| | IR | 0.73 | 0.54 | 0.60 | 0.79 | 0.73 | 0.69 | .66 | 0.77 | 0.82 | 0.83 | 0.85 | 3.8* (1.4, 6.3) | 0.006 |
| | RF (%) | 11.7 | 8.5 | 7.9 | 13.5 | 12.3 | 11.2 | 10.6 | 12.5 | 12.7 | 14.0 | 13.4 | 4.2* (0.6, 7.9) | 0.028 |
| **whites** | | | | | | | | | | | | | | |
| Total | Cases | 24517 | 2609 | 2525 | 2574 | 2391 | 2357 | 2322 | 2522 | 2430 | 2471 | 2316 | | |
| | IR | 3.03 | 3.42 | 3.26 | 3.30 | 3.02 | 2.96 | 2.84 | 3.01 | 2.92 | 2.92 | 2.74 | −2.1* (−2.9, −1.2) | 0.001 |
| | RF (%) | 11.5 | 12.9 | 12.4 | 12.2 | 11.4 | 11.0 | 10.8 | 11.4 | 10.9 | 11.3 | 10.7 | −3.8* (−6.6, −0.9) (Year 08–12) −0.2 (−2.4, 2.0) (Year 12–17) | 0.021 / 0.796 |
| Males | Cases | 12486 | 1320 | 1239 | 1297 | 1179 | 1210 | 1197 | 1330 | 1264 | 1251 | 1199 | | |
| | IR | 3.30 | 3.75 | 3.44 | 3.58 | 3.20 | 3.21 | 3.09 | 3.37 | 3.22 | 3.18 | 3.00 | −1.8* (−2.9, −0.7) | 0.006 |
| | RF (%) | 10.2 | 11.6 | 10.7 | 10.9 | 9.9 | 9.8 | 9.6 | 10.4 | 9.8 | 9.8 | 9.6 | −1.6* (−2.6, −0.6) | 0.007 |
| Females | Cases | 12031 | 1289 | 1286 | 1277 | 1212 | 1147 | 1125 | 1192 | 1166 | 1220 | 1117 | | |
| | IR | 2.80 | 3.11 | 3.10 | 3.05 | 2.87 | 2.75 | 2.62 | 2.70 | 2.66 | 2.68 | 2.50 | −2.4* (−3.1, −1.7) | <0.001 |
| | RF (%) | 13.2 | 14.6 | 14.8 | 13.9 | 13.4 | 12.7 | 12.4 | 12.9 | 12.4 | 13.2 | 12.2 | −1.9* (−2.8, −0.9) | 0.002 |
| **blacks** | | | | | | | | | | | | | | |
| Total | Cases | 1494 | 135 | 157 | 150 | 139 | 154 | 132 | 151 | 143 | 181 | 152 | | |
| | IR | 1.24 | 1.29 | 1.44 | 1.34 | 1.22 | 1.29 | 1.07 | 1.20 | 1.10 | 1.36 | 1.14 | −1.5 (−3.7, 0.7) | 0.152 |
| | RF (%) | 4.9 | 5.3 | 5.7 | 5.3 | 4.7 | 4.9 | 4.2 | 4.8 | 4.5 | 5.3 | 4.6 | −1.6 (−3.7, 0.6) | 0.137 |
| Males | Cases | 716 | 58 | 68 | 76 | 67 | 77 | 55 | 85 | 74 | 81 | 75 | | |
| | IR | 1.35 | 1.26 | 1.38 | 1.51 | 1.31 | 1.48 | 0.98 | 1.56 | 1.31 | 1.39 | 1.26 | −0.3 (−3.6, 3.0) | 0.816 |
| | RF (%) | 4.4 | 4.3 | 4.7 | 5.0 | 4.4 | 4.6 | 3.3 | 5.0 | 4.3 | 4.6 | 4.3 | −0.5 (−3.4, 2.5) | 0.709 |
| Females | Cases | 778 | 77 | 89 | 74 | 72 | 77 | 77 | 66 | 69 | 100 | 77 | | |
| | IR | 1.15 | 1.29 | 1.46 | 1.21 | 1.12 | 1.14 | 1.14 | 0.92 | 0.92 | 1.37 | 1.03 | −2.4 (−5.8, 1.2) | 0.163 |
| | RF (%) | 5.4 | 6.4 | 6.7 | 5.8 | 5.1 | 5.3 | 5.1 | 4.6 | 4.6 | 6.1 | 4.9 | −2.5 (−5.3, 0.4) | 0.079 |
| **Hispanics** | | | | | | | | | | | | | | |
| Total | Cases | 3956 | 336 | 304 | 397 | 381 | 390 | 405 | 403 | 435 | 435 | 470 | | |
| | IR | 2.34 | 2.45 | 2.10 | 2.68 | 2.39 | 2.36 | 2.37 | 2.24 | 2.33 | 2.23 | 2.32 | −0.7 (−2.3, 0.9) | 0.346 |
| | RF (%) | 10.5 | 10.7 | 9.4 | 11.5 | 10.7 | 10.6 | 10.8 | 10.0 | 10.6 | 10.1 | 10.9 | −0.1 (−1.5, 1.4) | 0.928 |
| Males | Cases | 1848 | 141 | 139 | 196 | 180 | 188 | 191 | 173 | 213 | 204 | 223 | | |
| | IR | 2.35 | 2.24 | 2.07 | 2.88 | 2.45 | 2.41 | 2.44 | 2.07 | 2.41 | 2.22 | 2.36 | −0.5 (−3.0, 2.1) | 0.658 |
| | RF (%) | 9.0 | 8.3 | 8.1 | 10.2 | 9.4 | 9.2 | 9.1 | 8.0 | 9.8 | 8.6 | 9.6 | 0.5 (−1.7, 2.6) | 0.636 |

*(Continued)*

**Table 1.** (Continued)

| | | Total | 2008 | 2009 | 2010 | 2011 | Year 2012 | 2013 | 2014 | 2015 | 2016 | 2017 | APC (95% CI) | p |
|---|---|---|---|---|---|---|---|---|---|---|---|---|---|---|
| Females | Cases | 2108 | 195 | 165 | 201 | 201 | 202 | 214 | 230 | 222 | 231 | 247 | | |
| | IR | 2.35 | 2.67 | 2.12 | 2.53 | 2.35 | 2.31 | 2.34 | 2.40 | 2.26 | 2.26 | 2.31 | −0.9 (−2.3, 0.6) | 0.197 |
| | RF (%) | 12.3 | 13.4 | 10.9 | 13.2 | 12.3 | 12.3 | 13.1 | 12.1 | 11.6 | 11.8 | 12.5 | −0.5 (−2.1, 1.1) | 0.466 |
| APIs | | | | | | | | | | | | | | |
| Total | Cases | 1600 | 149 | 119 | 132 | 152 | 148 | 165 | 185 | 160 | 196 | 194 | | |
| | IR | 1.35 | 1.52 | 1.17 | 1.22 | 1.35 | 1.28 | 1.38 | 1.51 | 1.22 | 1.49 | 1.38 | 0.7 (−1.7, 3.2) | 0.699 |
| | RF (%) | 8.9 | 9.8 | 7.7 | 8.1 | 8.8 | 8.5 | 9.2 | 9.5 | 8.0 | 9.5 | 9.5 | 0.9 (−1.2, 3.0) | 0.361 |
| Males | Cases | 826 | 84 | 65 | 63 | 81 | 76 | 86 | 95 | 82 | 96 | 98 | | |
| | IR | 1.56 | 1.89 | 1.43 | 1.32 | 1.63 | 1.46 | 1.60 | 1.74 | 1.40 | 1.61 | 1.57 | −0.3 (−3.0, 2.5) | 0.797 |
| | RF (%) | 8.4 | 10.0 | 7.8 | 7.1 | 8.7 | 7.6 | 8.8 | 8.9 | 7.5 | 8.7 | 8.8 | −0.1 (−2.8, 2.6) | 0.923 |
| Females | Cases | 774 | 65 | 54 | 69 | 71 | 72 | 79 | 90 | 78 | 100 | 96 | | |
| | IR | 1.18 | 1.22 | 0.98 | 1.13 | 1.12 | 1.14 | 1.20 | 1.30 | 1.07 | 1.39 | 1.23 | 1.7 (−0.6, 4.1) | 0.125 |
| | RF (%) | 9.5 | 9.4 | 7.6 | 9.2 | 9.0 | 9.7 | 9.8 | 10.1 | 8.7 | 10.4 | 10.4 | 1.9 (−0.1, 3.9) | 0.057 |

IR, World age-adjusted incidence rate; RF, relative frequency of FL among the incident NHL cases; APC, annual percent change; CI, confidence interval; TWs, Taiwanese; whites, non-Hispanic whites; blacks, non-Hispanic blacks; Hispanics, Hispanic whites; APIs, non-Hispanic Asians or Pacific Islanders. *The APC is significantly different from 0 at the alpha level of 0.05.

annual report was defined by ICD-O-3 codes 9597/3, 9675/3, 9690/3, 9691/3, 9695/3, and 9698/3, that is different from the definition of FL in SEER. Although we could not know the number of cases in each ICD-O-3 codes, the ICD-O-3 code 9675/3 is not applicable since 2010. The cases with code 9597/3 were very rare in Taiwan, just one case was found in a medical center from January 1998 to December 2014 [43] and there were no case in other medical center between January 2001 and December 2010 in Taiwan [44], thus it would not affect the results in this study.

In summary, this study found that the incidence of FL increased from 2008 to 2017 in Taiwan but decreased in the whites while remaining stable in the blacks, Hispanics, and APIs in the US during the same period. The patterns were similar between the males and females. A similar upward trend was also observed in Japan and South Korea. Future etiological investigations, particularly on modifiable risk factors, in Asian countries are warranted to understand the determinants of the upward trend of the incidence of FL.

## Supporting information

**S1 File.**
(DOCX)

**S1 Fig. Trends in the age-standardized incidence rates of follicular lymphoma in Taiwan, Japan, and Korea.**
(TIF)

## Acknowledgments

The authors would like to thank Enago (www.enago.tw) for the English language review.

## Author Contributions

**Conceptualization:** Yu-Chieh Su, Sheng-Fung Lin.

**Data curation:** Hung-Ju Li.

**Formal analysis:** Brian Chih-Hung Chiu, Hung-Ju Li.

**Investigation:** Wen-Chi Yang, Tsai-Yun Chen, Su-Peng Yeh, Ming-Chung Wang, Wen-Tsung Huang, Ming-Yang Lee.

**Methodology:** Yu-Chieh Su, Brian Chih-Hung Chiu, Hung-Ju Li.

**Supervision:** Yu-Chieh Su, Sheng-Fung Lin.

**Validation:** Wen-Chi Yang, Tsai-Yun Chen, Su-Peng Yeh, Ming-Chung Wang, Wen-Tsung Huang, Ming-Yang Lee.

**Writing – original draft:** Yu-Chieh Su, Brian Chih-Hung Chiu, Hung-Ju Li.

**Writing – review & editing:** Yu-Chieh Su, Brian Chih-Hung Chiu, Hung-Ju Li, Wen-Chi Yang, Tsai-Yun Chen, Su-Peng Yeh, Ming-Chung Wang, Wen-Tsung Huang, Ming-Yang Lee, Sheng-Fung Lin.

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
