## [Decision Letter · Decision Letter 0]

27 Aug 2021

PONE-D-21-21835

Investigation of the incidence trend of follicular lymphoma from 2008 to 2017 in Taiwan and the United States using population-based data

PLOS ONE

Dear Dr. Su,

Thank you for submitting your manuscript to PLOS ONE. After careful consideration, we feel that it has merit but does not fully meet PLOS ONE’s publication criteria as it currently stands. Therefore, we invite you to submit a revised version of the manuscript that addresses the points raised during the review process.

Specifically please identify the potential limitations and biases/confounding factors ,if they affect study results and measures to control them as this is essential for the practice of evidence-based medicine.

We look forward to receiving your revised manuscript.

Kind regards,

Amir Radfar, MD,MPH,MSc,DHSc

Academic Editor

PLOS ONE

2. In your ethics statement in the manuscript and in the online submission form, please provide additional information about the patient records used in your retrospective study. Specifically, please ensure that you have discussed whether all data were fully anonymized before you accessed them.

4. Please amend your list of authors on the manuscript to ensure that each author is linked to an affiliation. Authors’ affiliations should reflect the institution where the work was done (if authors moved subsequently, you can also list the new affiliation stating “current affiliation:….” as necessary)."

Reviewers' comments:

Reviewer's Responses to Questions

**Comments to the Author**

1. Is the manuscript technically sound, and do the data support the conclusions?

Reviewer #1: Partly

Reviewer #2: Yes

2. Has the statistical analysis been performed appropriately and rigorously? 

Reviewer #1: Yes

Reviewer #2: I Don't Know

3. Have the authors made all data underlying the findings in their manuscript fully available?

Reviewer #1: Yes

Reviewer #2: Yes

4. Is the manuscript presented in an intelligible fashion and written in standard English?

Reviewer #1: Yes

Reviewer #2: Yes

5. Review Comments to the Author

Reviewer #1: For people working in cancer research, an article like this is highly valuable, because the results allow to appraise and to put into context epidemiologic as well as clinical research in this area. In addition, those data are required, if data from different countries are jointly analysed in international collaborations. Furthermore, such data are required for planning future studies on follicular lymphoma in Taiwan and other countries in Asia.

The statistical analysis is, to my knowledge, state of the art. However, the authors would improve the value of the manuscript by taking a deeper look into apparent differences in the annual percentage change between males and females as well as the analysis of age at diagnosis. The latter has not been addressed so far.

Item 1. Is the manuscript technically sound, and do the data support the conclusions?

1.

The results presented in Table 1 and Table S2 show apparent differences between males and females across the different populations. As exploratory analysis, statistical tests could be performed. For example, the difference of -1.8 in the APC between males and females in the Taiwanese population in Table S2 reveals a p-value p = 0.023, indicating a difference in the APC between males and females. This finding is in contrast to the statement that the temporal trends were similar. However, due to the explorative nature of these post-hoc analyses as well as the problem of multiple testing, possible findings have to be interpreted with caution.

2.

The SEER data show some differences across groups defined by age at diagnosis. For example, for all ethnicities, an APC of -1.9 was observed overall, while the estimates for the age groups were 8.6 for ages 0-19, -2.1 for ages 20-64 and -1.8 for ages ≥ 65. Estimates are also available for males and females. The variable age at diagnosis should also be available for the Taiwanese registry data. If so, the authors should add the estimates, as supplementary data. If data for groups by age

at diagnosis are not available, please add a note on this issue.

3.

In table S2, results from joint analysis with data from Ko et al. are presented. How were the data for Table S2 merged in order to estimate the APC? Moreover, Ko et al. count lymphoma with ICD-O-3 codes 95973 and 96753 as follicular lymphoma (Ko et al., Table 1). Did the authors consider how this discrepancy between this and their own definition of follicular lymphoma could influence the results? The authors could, for example, present the number of patients with ICD-O-3 codes 95973 and 96753 in their own data set.

Minor

1. Please provide the formal definition of the annual percentage change.

2. Please use the term “relative frequency of FL among the incident NHL cases” throughout the manuscript and in the tables and figures.

3. Please add the confidence intervals as well as the p-values to the last column of Table 1.

4. Caption to Table S2: reference number for Ko et al. should be 10 instead of 8.

5. Introduction: add the unit (per year) to the statement “incidence rate of 3·5 per 100,000 persons in 2008–2017”

6. Materials and Methods, page 7: change “race” to “race/ethnicity” in the sentence “The SEER program records race as assigned by the North American Association of Central Cancer Registries.”

7. Results, page 10: the – declining – estimates for APC in whites, all as well as males and females, should be negative, i.e. -2.1, -1.8 and -2.4.

8. Caption for figure 1: add “incidence” to the word rate.

Reviewer #2: (I) Summary

The authors of this manuscript explore recent trends in follicular lymphoma (FL) in Taiwan and United States (US) using cancer registry (Taiwan Cancer Registry and US SEER) data from 2008 to 2017 with the aim of elucidating incidence patterns in both populations. They provide recent data from both cancer registries on age-adjusted incidence rates, relative frequencies and annual percent changes (APC) of FL stratified by sex and race/ethnicity. Collected data show a steadily rising trend in FL incidence in Taiwan (+39% from 2008-17) as compared to declining incidences in US white population (-20% from 2008-17) and steady trend in Hispanic, black and Asian/Pacific Islander population. The authors discuss possible factors for changes in incidence (genetics, lifestyle, environment) and compared their newly gathered data on FL incidence to recent data from other Asian countries noting some similarities. They conclude that further etiological investigations in Asian countries are needed.

(II) Discussion of specific areas of improvement

1.) Abstract

- clear and concise, no comment

2.) Introduction

- no comment

3.) Materials and methods

- the US are ethnically quite diverse and how SEER handles ethnicity is explained in great detail, please add a short comment if ethnicity is an issue for the Taiwan Cancer Registry.

4.) Results

- 2nd paragraph, line 3: APCs and 95% CI are listed in Figures 1 and 2, while Table 1 provide only APCs, this should be corrected accordingly.

-2nd paragraph, line 5: data of % change is provided for FL incidence in Taiwan (+39%), but no similar data provided for decrease in US white population, consider adding data.

- S1 table: reference to study by Ko et al is 10 and not 8, please correct

- S1 table: data for Korea12 and 15 shown with same color and symbol making it harder to read, consider correcting

- Table 1, whites, RF (%), APC: only variable with a joint point. Discuss with your statistician if an average APC could be calculated and added in the comment under the table or consider adding a comment why there are two number as opposed to all other APCs.

5.) Discussion

- list possible limitations of this study and/or biases (for example: SEER data do not cover whole US population; incidence change in US white population cannot be generalized

(III) Other comments

- very fluent reading, carefully edited text, no language issues.

6. PLOS authors have the option to publish the peer review history of their article (what does this mean?). If published, this will include your full peer review and any attached files.

Reviewer #1: No

Reviewer #2: No

---

## [Author Response · Author response to Decision Letter 0]

7 Oct 2021

Revierwer #1:

Question 1.

The results presented in Table 1 and Table S2 show apparent differences between males and females across the different populations. As exploratory analysis, statistical tests could be performed. For example, the difference of -1.8 in the APC between males and females in the Taiwanese population in Table S2 reveals a p-value p = 0.023, indicating a difference in the APC between males and females. This finding is in contrast to the statement that the temporal trends were similar. However, due to the explorative nature of these post-hoc analyses as well as the problem of multiple testing, possible findings have to be interpreted with caution.

Answer: Thank you for your valuable opinions. We have provided it and marked it. (page 9, 10)

page 9: (Materials and Methods) The trend between males and females was compared by using a test of parallelism.

page 10: (Results) The tests of parallelism for the male and female trend lines in the Taiwanese population and US white population both were not significant (p = 0.456 and 0.133).

Question 2.

The SEER data show some differences across groups defined by age at diagnosis. For example, for all ethnicities, an APC of -1.9 was observed overall, while the estimates for the age groups were 8.6 for ages 0-19, -2.1 for ages 20-64 and -1.8 for ages ≥ 65. Estimates are also available for males and females. The variable age at diagnosis should also be available for the Taiwanese registry data. If so, the authors should add the estimates, as supplementary data. If data for groups by age at diagnosis are not available, please add a note on this issue.

Answer: The estimates had added in supplementary data, S3 Table. The APCs were 4.4 for ages 0-34, 2.6 for ages 35-64 and 4.0 for ages ≥ 65. For males, the APCs were 1.6, 2.7 and 2.9 for ages 0-34, 35-64, and ≥ 65, respectively. For females, the APCs were 8.2, 2.5 and 5.5 for ages 0-34, 35-64, and ≥ 65, respectively.

 Because the numbers of FL cases were almost zero for ages < 30 from 2008 to 2017, we stratified the cases into 3 age groups: 0-34, 35-64, and ≥ 65.

Question 3.

In table S2, results from joint analysis with data from Ko et al. are presented. How were the data for Table S2 merged in order to estimate the APC? Moreover, Ko et al. count lymphoma with ICD-O-3 codes 95973 and 96753 as follicular lymphoma (Ko et al., Table 1). Did the authors consider how this discrepancy between this and their own definition of follicular lymphoma could influence the results? The authors could, for example, present the number of patients with ICD-O-3 codes 95973 and 96753 in their own data set.

Answer: The age-standardized incidence rates (ASR) between 2002 and 2007 from Ko et al. (Table 2) and the ASR between 2008 and 2017 from our data were merged to estimate the APC by using the Jointpoint regression under the “constant variance assumption”. 

 The ICD-O-3 code 9675/3 is not applicable since 2010. The cases with code 9597/3 were very rare in Taiwan, just one case was found in a medical center from January 1998 to December 2014 [1] and there were no case in other medical center between January 2001 and December 2010 in Taiwan [2], thus it would not affect the results in this study.

Reference:

1. Liu KL, Tsai WC, Lee CH. Non-mycosis fungoides cutaneous lymphomas in a referral center in Taiwan: A retrospective case series and literature review. PLoS One. 2020;15(1): e0228046. doi: 10.1371/journal.pone.0228046. PMID: 31978091.

2. Lee CN, Hsu CK, Chang KC, Wu CL, Chen TY, Lee JY. Cutaneous lymphomas in Taiwan: A review of 118 cases from a medical center in southern Taiwan. Dermatologica sinica. 2018; 36(1): 16-24. Doi: 10.1016/j.dsi.2017.08.004.

Minor Questions:

1. Please provide the formal definition of the annual percentage change. 

Answer: We have provided it and marked it.

Page 8: APC is often used to characterize trends in disease rates, which assume the disease rates change at a constant percentage of the rate of the previous year by using this approach.

2. Please use the term “relative frequency of FL among the incident NHL cases” throughout the manuscript and in the tables and figures.

Answer: We have corrected it and marked it throughout the manuscript and in the tables and figures.

3. Please add the confidence intervals as well as the p-values to the last column of Table 1.

Answer: We have corrected it and marked it.

4. Caption to Table S2: reference number for Ko et al. should be 10 instead of 8.

Answer: We have corrected it and marked it for reference number in Supporting Information.

5. Introduction: add the unit (per year) to the statement “incidence rate of 3·5 per 100,000 persons in 2008–2017”

Answer: We have corrected it and marked i: incidence rate of 3·5 per 100,000 persons per year in 2008–2017

6. Materials and Methods, page 7: change “race” to “race/ethnicity” in the sentence “The SEER program records race as assigned by the North American Association of Central Cancer Registries.”

Answer: We have corrected it and marked it. (Page 3, 7, 9)

7. Results, page 10: the – declining – estimates for APC in whites, all as well as males and females, should be negative, i.e. -2.1, -1.8 and -2.4.

Answer: We have corrected it and marked it.

8. Caption for figure 1: add “incidence” to the word rate.

Answer: We have corrected it and marked it.

 

Revierwer #2:

Reviewer #2: (I) Summary

The authors of this manuscript explore recent trends in follicular lymphoma (FL) in Taiwan and United States (US) using cancer registry (Taiwan Cancer Registry and US SEER) data from 2008 to 2017 with the aim of elucidating incidence patterns in both populations. They provide recent data from both cancer registries on age-adjusted incidence rates, relative frequencies and annual percent changes (APC) of FL stratified by sex and race/ethnicity. Collected data show a steadily rising trend in FL incidence in Taiwan (+39% from 2008-17) as compared to declining incidences in US white population (-20% from 2008-17) and steady trend in Hispanic, black and Asian/Pacific Islander population. The authors discuss possible factors for changes in incidence (genetics, lifestyle, environment) and compared their newly gathered data on FL incidence to recent data from other Asian countries noting some similarities. They conclude that further etiological investigations in Asian countries are needed.

Answer: Thank you for your comment.

(II) Discussion of specific areas of improvement

1.) Abstract

- clear and concise, no comment

2.) Introduction

- no comment

3.) Materials and methods

- the US are ethnically quite diverse and how SEER handles ethnicity is explained in great detail, please add a short comment if ethnicity is an issue for the Taiwan Cancer Registry.

Answer: Individual data collected by Taiwan Cancer Registry doesn’t include patients’ ethnicity.

4.) Results

- 2nd paragraph, line 3: APCs and 95% CI are listed in Figures 1 and 2, while Table 1 provide only APCs, this should be corrected accordingly.

Answer: We have corrected it and marked it in Table 1. 

-2nd paragraph, line 5: data of % change is provided for FL incidence in Taiwan (+39%), but no similar data provided for decrease in US white population, consider adding data.

Answer: We have provided it and marked it in 2nd paragraph, line 12: from 3·42 to 2·74 per 100,000 person-years between 2008 and 2017, representing a 20% decrease.

- S1 table: reference to study by Ko et al is 10 and not 8, please correct

Answer: We have corrected it and marked it for reference number in Supporting Information.

- S1 table: data for Korea12 and 15 shown with same color and symbol making it harder to read, consider correcting

Answer: We have corrected it and marked it: Korea_Lee, Korea_Kim.

- Table 1, whites, RF (%), APC: only variable with a joint point. Discuss with your statistician if an average APC could be calculated and added in the comment under the table or consider adding a comment why there are two number as opposed to all other APCs.

Answer: We have corrected it and marked it. 

Page 17: In the whites, the relative frequency of FL among the incident NHL cases declined from 12.9% to 10.7%, the average APC was −1.8 (95% CI: −3.1% to −0.5%, p = 0.009).

5.) Discussion

- list possible limitations of this study and/or biases (for example: SEER data do not cover whole US population; incidence change in US white population cannot be generalized

Answer:

 We have corrected it and marked it.

 Page 21: The main limitation of this study is that SEER data does not cover whole US population, the incidence change in each racial population in US cannot be generalized. Another limitation of our study is that the FL data in Taiwan was retrieved from the TCR annual report, not Taiwan Cancer Registry Database. The number of FL cases by gender and age in the TCR annual report was defined by ICD-O-3 codes 9597/3, 9675/3, 9690/3, 9691/3, 9695/3, and 9698/3, that is different from the definition of FL in SEER. Although we could not know the number of cases in each ICD-O-3 codes, the ICD-O-3 code 9675/3 is not applicable since 2010. The cases with code 9597/3 were very rare in Taiwan, just one case was found in a medical center from January 1998 to December 2014 [41] and there were no case in other medical center between January 2001 and December 2010 in Taiwan [42], thus it would not affect the results in this study.

(III) Other comments

- very fluent reading, carefully edited text, no language issues.

Answer: Thanks for your comments.

---

## [Decision Letter · Decision Letter 1]

7 Jan 2022

PONE-D-21-21835R1Investigation of the incidence trend of follicular lymphoma from 2008 to 2017 in Taiwan and the United States using population-based dataPLOS ONE

Dear Dr. Su,

Thank you for submitting your manuscript to PLOS ONE. After careful consideration, we feel that it has merit but does not fully meet PLOS ONE’s publication criteria as it currently stands. Therefore, we invite you to submit a revised version of the manuscript that addresses the points raised during the review process.

Specifically,Please make sure that the limitation section of your paper indicates" that despite the SEER database, the Taiwan Cancer registry doesn’t include patients’ ethnicity".Please explain how the study handled missing data.Please answer comments made by reviewer#3

We look forward to receiving your revised manuscript.

Kind regards,

Amir Radfar, MD,MPH,MSc,DHSc

Academic Editor

PLOS ONE

Reviewers' comments:

Reviewer's Responses to Questions

**Comments to the Author**

1. If the authors have adequately addressed your comments raised in a previous round of review and you feel that this manuscript is now acceptable for publication, you may indicate that here to bypass the “Comments to the Author” section, enter your conflict of interest statement in the “Confidential to Editor” section, and submit your "Accept" recommendation.

Reviewer #2: (No Response)

Reviewer #3: (No Response)

2. Is the manuscript technically sound, and do the data support the conclusions?

Reviewer #2: Yes

Reviewer #3: Partly

3. Has the statistical analysis been performed appropriately and rigorously? 

Reviewer #2: I Don't Know

Reviewer #3: No

4. Have the authors made all data underlying the findings in their manuscript fully available?

Reviewer #2: Yes

Reviewer #3: Yes

5. Is the manuscript presented in an intelligible fashion and written in standard English?

Reviewer #2: Yes

Reviewer #3: Yes

6. Review Comments to the Author

Reviewer #2: The authors have adequately addressed all questions from the previous round of review posted by both reviewers. The manuscript has been improved in the Materials, Results and Discussion sections, tables and charts have been revised according to recommendations. At this point I have no further comments.

Reviewer #3: 1- Why the authors compared the incidence of FL in Taiwan and US? What is the rationale behind it?

2- It is highly recommended to provide the report based on the guidelines such as STROBE or the following: https://www.ncbi.nlm.nih.gov/books/NBK208602/.

The authors should provide some subheadings in the methods to cover the main following questions regarding registry data:

• Study purpose: Were the objectives/hypotheses predefined or post hoc?

• Patient population: Who was studied?

• Data quality: How were the data collected, reviewed, and verified?

• Data completeness: How were missing data handled?

• Data analysis: How were the analyses chosen and performed?

3- In the abstract the authors stated that “Our findings suggest that modifiable risk factors may be important determinants of the diverse trend of FL, especially in Taiwan.” This conclusion does not seem logical since the authors did not examine the role of modifiable risk factors in this study. Conclusions should be consistent with the findings of the study.

4- It is recommended to quantify the association of age, period, and cohort using age–period–cohort models.

5- Reporting incidence rate ratios (IRRs) for assess the cohort effect is recommended.

7. PLOS authors have the option to publish the peer review history of their article (what does this mean?). If published, this will include your full peer review and any attached files.

Reviewer #2: No

Reviewer #3: No

---

## [Author Response · Author response to Decision Letter 1]

20 Jan 2022

Answer to Editor’s comments:

• Please make sure that the limitation section of your paper indicates" that despite the SEER database, the Taiwan Cancer registry doesn’t include patients’ ethnicity".

• Please explain how the study handled missing data.

• Please answer comments made by reviewer#3

Answer: Thank you for your reminder. 

1. The TCR annual report does not include patients’ ethnicity have been confirmed. 

2. The follicular lymphoma data in Taiwan and US available from the Taiwan Cancer Registry annual report of the Health Promotion Administration, Ministry of Health and Welfare, Taiwan and the US National Cancer Institute’s SEER 18 program, respectively. The cancer registry annual report based on the databased established by the Taiwan Cancer Registry which case completeness rates greater than 97%. The Taiwan Cancer Registry implemented rigorous quality control protocols to validate and verify cancer diagnoses, including submission of tissue pathology reports and random sampling of claims and diagnosis reports. It is considered a complete and accurate registry. The SEER program records race as assigned by the North American Association of Central Cancer Registries, which is only comprehensive source of population-based in the US and have a case completeness rate greater than 98%.

Answer to Reviewers’ comments:

Revierwer #2: The authors have adequately addressed all questions from the previous round of review posted by both reviewers. The manuscript has been improved in the Materials, Results and Discussion sections, tables and charts have been revised according to recommendations. At this point I have no further comments.

Answer: Thank you for your comments. 

 

Revierwer #3:

1- Why the authors compared the incidence of FL in Taiwan and US? What is the rationale behind it?

Answer: We want to compare the trends for incidence rates of follicular lymphoma in Taiwan and other Asian ethnic groups in recent years, but we cannot obtain these data from Japan, South Korea and other Asian countries. Therefore, we compared the data for Taiwanese and Asian ethnic groups in US from SEER database, and also observed the trends for incidence rates of FL for other ethnic groups in US from SEER database.

2- It is highly recommended to provide the report based on the guidelines such as STROBE or the following: https://www.ncbi.nlm.nih.gov/books/NBK208602/.

The authors should provide some subheadings in the methods to cover the main following questions regarding registry data:

• Study purpose: Were the objectives/hypotheses predefined or post hoc?

• Patient population: Who was studied?

• Data quality: How were the data collected, reviewed, and verified?

• Data completeness: How were missing data handled?

• Data analysis: How were the analyses chosen and performed?

Answer: Thank you for your comments. We have provided it and marked it in red (Page 6-9).

3- In the abstract the authors stated that “Our findings suggest that modifiable risk factors may be important determinants of the diverse trend of FL, especially in Taiwan.” This conclusion does not seem logical since the authors did not examine the role of modifiable risk factors in this study. Conclusions should be consistent with the findings of the study.

Answer: Thank you for your comments. We have deleted this conclusion in the abstract.

4- It is recommended to quantify the association of age, period, and cohort using age–period–cohort models.

5- Reporting incidence rate ratios (IRRs) for assess the cohort effect is recommended.

Answer: Thank you for your comments. Since the number of FL cases in Taiwan were showed by age in 5-year age groups from the TCR annual report and the study period was from 2008 to 2017, the number of consecutive 5-year periods was just two, therefore, the age-period-cohort model was not applicable. We will use the age-period-cohort models in the future studies.

---

## [Decision Letter · Decision Letter 2]

4 Mar 2022

Investigation of the incidence trend of follicular lymphoma from 2008 to 2017 in Taiwan and the United States using population-based data

PONE-D-21-21835R2

Dear Dr. Su,

We’re pleased to inform you that your manuscript has been judged scientifically suitable for publication and will be formally accepted for publication once it meets all outstanding technical requirements.

Kind regards,

Carla Pegoraro

Staff Editor

PLOS ONE

Additional Editor Comments (optional):

Reviewers' comments:

Reviewer's Responses to Questions

**Comments to the Author**

1. If the authors have adequately addressed your comments raised in a previous round of review and you feel that this manuscript is now acceptable for publication, you may indicate that here to bypass the “Comments to the Author” section, enter your conflict of interest statement in the “Confidential to Editor” section, and submit your "Accept" recommendation.

Reviewer #2: All comments have been addressed

Reviewer #3: All comments have been addressed

2. Is the manuscript technically sound, and do the data support the conclusions?

Reviewer #2: Yes

Reviewer #3: Yes

3. Has the statistical analysis been performed appropriately and rigorously? 

Reviewer #2: Yes

Reviewer #3: Yes

4. Have the authors made all data underlying the findings in their manuscript fully available?

Reviewer #2: Yes

Reviewer #3: Yes

5. Is the manuscript presented in an intelligible fashion and written in standard English?

Reviewer #2: Yes

Reviewer #3: Yes

6. Review Comments to the Author

Reviewer #2: The authors have addressed questions from the previous round of review. The manuscript has been adequately improved according to STROBE guidelines and presents data in a clear way. No further comments.

Reviewer #3: Dear editor

I reviewed the manuscript and all comments have been addressed adequately by authors.

regards

7. PLOS authors have the option to publish the peer review history of their article (what does this mean?). If published, this will include your full peer review and any attached files.

Reviewer #2: No

Reviewer #3: No

---

## [Editor Report · Acceptance letter]

8 Mar 2022

PONE-D-21-21835R2 

Investigation of the incidence trend of follicular lymphoma from 2008 to 2017 in Taiwan and the United States using population-based data 

Dear Dr. Su:

I'm pleased to inform you that your manuscript has been deemed suitable for publication in PLOS ONE. Congratulations! Your manuscript is now with our production department. 

Kind regards, 

on behalf of

Dr Carla Pegoraro 

Staff Editor

PLOS ONE